# Community Water Fluoridation: Caveats to Implement Justice in Public Oral Health

**DOI:** 10.3390/ijerph18052372

**Published:** 2021-03-01

**Authors:** Youngha Song, Junhewk Kim

**Affiliations:** 1Department of Preventive and Social Dentistry, School of Dentistry, Seoul National University, Seoul 03080, Korea; youngha.song@adelaide.edu.au; 2Australian Research Centre for Population Oral Health, Adelaide Dental School, The University of Adelaide, Adelaide 5000, Australia; 3Department of Dental Education, College of Dentistry, Yonsei University, Seoul 03722, Korea

**Keywords:** public dentistry, oral prevention, community water fluoridation, public health ethics, procedural justice

## Abstract

Community water fluoridation (CWF), a long-established public health intervention, has been studied for scientific evidence from both of yea and nay standpoints. To justify CWF with scientific evidence inevitably leads to ethical justification, which raises the question of whether oral health is of individual concern or social responsibility. As dental caries is a public health problem, public health ethics should be applied to the topic instead of generic clinical ethics. From both pro- and anti-fluoridationists’ perspectives, CWF is a public health policy requiring a significant level of intervention. Thus, there needs to take further considerations for justifying CWF beyond the simple aspect of utility. For further ethical considerations on CWF, three caveats were suggested: procedural justice, social contexts, and maintenance of trust. The process to justify CWF should also be justified, not simply by majority rule but participatory decision-making with transparency and pluralistic democracy. Social contexts are to be part of the process of resolving conflicting values in public health interventions. Public trust in the dental profession and the oral healthcare system should be maintained over the considerations. This article suggests accountability for reasonableness as a framework to consider infringement by CWF for public justification of its implementation.

## 1. Introduction: Rocket Science or Preponderance of Evidence?

Community water fluoridation (CWF) is a long-established [1] but still ethically contentious [2] public health scheme. For a scientific background on CWF, a large portion of empirical findings support its benefit of preventing dental caries, including systematic reviews [3,4,5]. However, there are also contrarian studies raising concerns about the safety [6,7] and efficacy of an “optimal” intake of fluoride [8]. Considering the importance of evidence-based healthcare, consistent/continuous accumulation of robust scientific evidence by refuting each other [9,10] for the justification of CWF seems required. If the frame of reference is set on philosophical and ethical considerations, however, the rationale for CWF is no longer empirically testable. For that matter, there remains a pragmatic issue of “acceptability” [11] to the various stakeholders of CWF.

Ethical claims have been primarily raised by bioethicists to point out the philosophical weaknesses of CWF [12,13]. Their two major arguments regarding its ethics can be summarised as the principle of non-maleficence and the breach of individual autonomy [13]. In CWF which aims to reduce dental caries among low-income populations, no specific issues have been raised about beneficence and justice. The agendas have been refuted by proponents of CWF with scientific evidence and ethical justification briefly as follows:*Non-maleficence*: reliable evidence suggesting that CWF causes possible harms and/or adverse events on health is rare, except dental fluorosis [3,5]. Even dental fluorosis can be minimised by a safe concentration of fluoride for less aesthetic and oral health concerns [5];*Autonomy*: CWF, as a public health intervention to protect the common good, the general public’s oral health and the reduction of health inequity can override individual freedom of choice and become exempt from seeking consent by a legitimate process of the representative system [5,14].

However, the refutations suggested are not sufficiently convincing for these contentions. For non-maleficence, whether the minimal grade of dental fluorosis is harmful or not remains debatable, leaving a further concern that the acceptance of possible harm should be approved by the individual’s choice, not the authority’s enforcement. Autonomy also leaves a remaining question about how to secure a proper agreement on the health policy for the unspecified general public, despite the possible exemption of seeking individual consent. Therefore, the agendas need to be dealt with in a different framework beyond the conventional refutation.

This article does not seek to reject or side with a certain claim on the matter of CWF. Instead of weighing up scientific findings or highlighting ethical justifications on CWF, we are trying to view the intervention from the perspective of public health ethics. The first aim of the paper is to localise where both pro- and anti-fluoridationists can agree in the discussion about CWF. The second is to suggest considerations for the ethical decision-making of CWF for better acceptability to the general public. Rather than devising a novel and effective ethical refutation of each other’s claim about CWF, caveats to lead a further discussion for the justification of CWF will be suggested. For the sake of argument on ethical issues in this paper, scientific evidence for both claims is set aside, since it is hard to find a middle ground between different interpretations of the utility of CWF. In other words, the utility of CWF is out of focus in the discussion of this paper, leaving it to be determined through more empirical studies. This is not because scientific evidence has little to do with ethical justification, but because we seek instead to concentrate on values and normativity regarding CWF.

## 2. Can CWF Be an Issue of Public Health Ethics?

Dental caries, as the main subject of CWF for public oral health, is a public health problem [15]. Before starting a discussion about public health interventions, it should be warranted first whether the health issue is classified as such. According to the criteria for a public health problem suggested by Sheiham et al. [16], dental caries satisfies all four conditions: high prevalence of the disease, significant impact on an individual level, considerable impact on wider society, and availability of prevention and effective treatment. From a more general definition suggested by Acheson and endorsed by the World Health Organization [17], dental caries should also be dealt with “through organized efforts of society” considering its explicit social gradients [18]. Henceforth, the coping strategy against dental caries in *public* oral health should be different from that in *clinical* dentistry. The banality of this common knowledge sometimes causes confusion in ethical considerations on health issues.

If dental caries is considered a public health problem, at least for the discussion of CWF, it is reasonable to apply *public health* ethics to the topic, not clinical ethics. Hitherto, it has not been uncommon to view public health problems with the values of clinical ethics [19]. That is because bioethics emerged from protecting patients in medical practice and participants in clinical research. The recognition of applying ethical values to public health and its importance for policymaking has arisen only as of late compared to clinical ethics.

Most distinctively, the “public” in public health has two prescriptive meanings: “the health of the public” and “collective interventions” [20]. In other words, despite being under the same superordinate concept of bioethics, public health ethics should differ from clinical ethics, which focuses on individual patients’ needs and concerns from healthcare practitioners and systems [19]. The details of the matter lie beyond the scope of this paper, but the contrast of core principles in each discipline can representatively exhibit the difference: autonomy, beneficence, non-maleficence, and justice as four principles for clinical/medical ethics [21]; and utility, liberty, and equality as three for public health ethics [22]. Liberty corresponds to autonomy as in public health policy for the general public and population group, and likewise equality to justice. However, what is the meaning and weight that “consent” in public health policy carries in contrast to clinical practice for individuals? For example, if a resident agrees on the health education project for the local community, to what extent and until when does the consent remain valid? Can the consent be expanded or assumed to a modified yet consistent policy with the same rationale? The core value of securing an individual’s informed consent in clinical ethics may need to be revisited for public health ethics.

For that matter, ethical frameworks have been suggested to resolve conflicts between values in public health interventions [23]. To produce benefits, prevent harms, and maximise utility, five justificatory conditions are demonstrated as follows: effectiveness, proportionality, necessity, least infringement, and public justification. An ethical evaluation of CWF by means of the framework was also attempted (Table 1). The table is a modified version of the framework suggested by Childress et al. [23] to clarify the ambiguous interpretation (necessity and least infringement are hard to distinguish) [24] for CWF. Among the proposed five conditions, the first three (effectiveness, proportionality, and necessity) have been sufficiently applied from the early stages of CWF (Table 1). However, scientific findings still remain contentious regarding the level of evidence from the standpoints of pro- and anti-CWF [25]. Instead of dealing with all conditions, the two latter conditions of least infringement and public justification are discussed for ethical considerations, as we expect the issue of infringement to be the point from which both parties share and start a debate. In the latter part of this paper, this point will lead to public justification, which is to be focused on in the section of “three caveats”.

## 3. Slanted Rungs in the Intervention Ladder

In order to better understand the infringement of individual liberty by public health interventions, let us refer to a schematic diagram. Despite its criticism for over-simplification in public health policymaking [28], the “Intervention Ladder” [29] is a useful metaphor representing the level of liberty infringement in public health interventions (Figure 1). The ladder has been utilised on several occasions for the demonstration of public health ethics and modification has been attempted to improve its unidimensional approach [28,30,31]. The model is easy to follow on the matter of liberty with the simple notion that the higher the rung is in the ladder, the more justified the intervention should be [30]. 

Proponents of CWF try to place it on the level tagged “restrict choice.” Once CWF is implemented, virtually no other options of drinking non-fluoridated community water are feasible. It is possible to choose bottled water for drinking instead of fluoridated community water, but the unavoidable cost for alternatives can be a restrictive factor for freedom of choice. Additionally, CWF should affect the consumption of other fluoride-containing products, e.g., fluoridated salt, which endorses its restrictive nature against an individual’s free choice. However, pro-fluoridationists support CWF as a less harmful public health intervention with a certain way of obtaining consent among others, despite its restriction of choice. Particularly, in public health approaches, the strength of the whole-population strategy can be demonstrated for the prevention of dental caries in the general public rather than the high-risk strategy [32]. Furthermore, CWF as a default choice is in accordance with the central concept of public health promotion “making the healthier choices the easier choices” [33]. For this reason, proponents argue that the restriction of choice can be outweighed by the public health benefits CWF brings to the general public.

On the other hand, CWF is deemed to be on the highest rung to the opponents. Anti-fluoridationists can feel that their constitutional right to freedom of choice is practically deprived by the enforcement of “artificial” water treatment. Furthermore, the “mass medication” of the general public [13] has not been legally consented to by those directly affected in health. Although the benefit of preventing dental caries by CWF can be admitted as a public health goal, unsolicited or unconsented interventions are considered intrusive paternalism [34] or nanny-state power [30]. Regarding the value of individual liberty, opponents put CWF on the top of the ladder, where the strongest justification is required for the enforcement of the intervention.

As previously mentioned, the same empirical findings from a scientific article [3] have been exemplified by both opposing arguments of pro- [14] and anti-fluoridationists [13]. To emphasise the effectiveness and necessity of CWF would not cross the chasm between two opponents, which is poles apart. Rather, the restrictive nature that CWF bears is what both parties can agree upon, as suggested with the issue of infringement in the previous section. Its nature does not constitute a rationale to dismiss CWF in its own right, as the public benefits CWF can bring should not be underestimated and restriction of choice is a demand for alternatives of least infringement, not a “deal-breaker” of public policy [24].

At present, what matters more about the ethical considerations of CWF highlights how to resolve the inherent restriction of choice. The resolution of the matter seems to reach a consensus that both proponents and opponents can co-occupy. We present three caveats to justify the infringement of public health policy: procedural justice, social context, and maintenance of trust. These caveats are derived from the effort to resolve the restriction of choice and minimise the infringement through public justification.

## 4. Three Caveats for Ethical Considerations on Community Water Fluoridation

### 4.1. Procedural Justice: The End Does Not Always Justify The Means

The process to justify public health interventions among conflicting values between individuals and society should also be justified. This is not a tautology but a less commonly illuminated condition compared with the contested ethical principles. When the benefits and burdens are fairly distributed (distributive justice), the result of fair distribution can be legitimate only insofar as public participation, including stakeholders, is secured (procedural justice) [23]. As is mentioned above, the underlying value of least infringement is to pursue the implementation to minimise the infringement of the public policy, and procedural justice enables making a decision that a community can accept through evaluating the diverse options available. For example, in the case of CWF, not only professionals in oral health expertise but also stakeholders with different views (e.g., bioethicists, public health advocates, human rights activists, and environmentalists) are to be encouraged to participate in the decision-making process. Transparency [35], as a key concept of public justification, plays a pivotal role in reaching a legitimate conclusion through pluralistic democracy [23]. Therefore, democratic decision-making [30] with transparency and participatory discussion is, by all means, the premise to resolving conflicts in values.

One pitfall that should be avoided is the attempt to finalise the issue simply by majority rule in the name of democracy. The *prima facie* fair rule has two fallacies for the justification of public health interventions: the tyranny of the majority [36] and analysis paralysis [37]. The first risk can commonly be applied when the accumulation of the majority’s less-significant utility takes precedence over the vital needs of the minority by justified oppression in casting ballots. This defies another important rule of libertarian egalitarianism, the “difference principle”: inequalities are permitted if they benefit the least well-off in society [38]. 

The other fallacy stems from the complexity [39] of social issues. In this modern society, the more contentious a social issue becomes, the more complex and sophisticated the supporting arguments for its refutation develop, and the harder it is for the general public to understand and make a decision. If average people are not eager for “informed” participation in the public health intervention, the decision made by simply voting on the issue would be nothing but a popularity contest about how compelling each group’s claims look. The role of stakeholders on CWF should be to provide the general public with comprehensive and approachable advice based on their professional responsibilities. To avoid the risks, politically and sociologically better-designed methods of decision-making are necessary, such as community juries [40], for the general public’s participation in a more intensive/encouraged form on the contextual level.

To secure procedural justice in CWF, we would suggest the framework of Accountability for Reasonableness proposed by Daniels and Sabin [41]. The framework introduces fair procedures in establishing a public health policy from four conditions: publicity of evidence, reasonable construal for public acceptance, development of dispute resolution mechanism, and enforcement of the three conditions in the procedure (Table 2) [42,43]. As accountability for reasonableness aims at fair procedures for conflicting values and rationales, CWF can also benefit from the framework to choose priorities among the competing principles proponents and opponents hold for CWF. For example, Accountability for Reasonableness can enable the fair procedure for CWF by making relevant resources/information publicly accessible to the stakeholders in the community, providing reasonable construal for the agreement of all fair-minded parties, and establishing the mechanism to raise concerns and revise the decision in case of further evidence.

### 4.2. Social Contexts: Comparing Apples and Oranges

For resolving conflicting values in public health interventions, social contexts should be part of the equation along with ethical principles. Selgelid suggests that moderate pluralism can be a fourth framework to strike a balance among the three principles in public health ethics when in conflict [22]. The framework helps place a public health policy within the range of an ethical spectrum rather than determining whether the policy can be adopted or not [44]. Social contexts are of central concern in moderate pluralism, as those conflicting values are apparently incommensurable without contextual considerations [22]. For instance, the amount of each value is calculated by the formula based on social contexts (e.g., proportionality as utility divided by liberty), which can enable decision-makers to consider how much each value should be reflected for the policy. 

More pragmatically, public attitudes towards public health interventions are reported to be different according to social contexts in an empirical study [45]. In this regard, it is reasonable to adapt the intervention to *local* contexts for better implementation, in spite of its established universal evidence base from academic studies [46]. However, attempts to selectively choose and apply relevant evidence out of context have been tried for the support/opposition of public health interventions (e.g., the exemplification of enforcement or revocation of CWF in other countries or communities without considering social background and contexts). If we consider Australia, South Korea, and the USA for a comparison of social contexts, they are as different as “apples and oranges.” To name a few differences regarding individual liberty and responsibility of public health, Australia and the USA take rather different stances. Australia has enforced a strict helmet law in cycling since 1990 to reduce bicycle-related head injury [47], whereas only 19 states in the USA uphold a universal helmet law on motorcycles, despite the drastic relapse of motorcycle-related fatalities compared with before the repeal of the law [48]. For fluorosis as an adverse effect of CWF, aesthetic considerations in Korea, where attention to cosmetics and appearances is of the highest concern [49], must be different from that of the generally lower concern in the Australian population [5]. For financial access to dental care services, South Korea supports universal healthcare insurance, including regular preventive dental services [50]. In contrast, healthcare reforms to expand medical coverage in the USA have prompted social and political debates [51], let alone dental services. For this reason, the decision-making on CWF in a society should be based on the consideration of contextual differences.

### 4.3. Maintenance of Trust: Good Intentions and Good Works

The final proviso for ethical considerations on CWF is the maintenance of trust in the dental profession and the oral healthcare system. By the nature of public health initiatives, CWF has been driven by oral health professionals [52], in spite of the occupational disadvantage that CWF may have led to a decrease in the number of dental establishments [53]. However, not all public health interventions initiated from goodwill are acknowledged by the general public as intended, and some even face untoward “backfire” from mutual misunderstanding. If CWF starts feeling coercive or enforced by authorities in either state power or the dental profession for professional prestige [54], it may cause antipathy over apathy accompanied by structured resistance, even with conspiracy theories [55]. The worst scenario of adverse events from the conflicts on CWF is being suspicious of the integrity in professionalism and subsequent loss of trust in the profession and system [55,56]. 

COVID-19 unveils the importance of trust in public health ethics in an unexpected and drastic manner [57,58]. Likewise, CWF, as a public health intervention, should be based on public trust in the dental profession and authorities of policy implementation. The World Health Organization and Community Preventive Services Task Force of CDC also suggest that the barriers to water fluoridation are social acceptability and media attention [59,60]; therefore, the policy should be based on social trust and transparent political strategies. 

Public trust can be initiated from the provision of transparent information and communication, to which the profession has hitherto insufficiently committed. Thus, maintaining trust and honesty, one of the commitments in the charter of medical professionalism [61], should be accepted and pursued by the dental profession through transparent communication, as the American Dental Association suggests veracity for ethics principles. To explain the benefits and harms CWF brings to the community and persuade them is the profession’s undertaking as one of the social responsibilities necessary for an informed decision of the general public. In other words, CWF could be a practical touchstone for the dental profession about how to establish public trust and communicate with society.

## 5. Conclusions: You Can Lead a Horse to Water, but You Cannot Make It Drink

To date, we have found that CWF needs to be justified in ethical considerations, and public health ethics is an adequate framework for the discussion. For the different recognition of CWF, the metaphor of the intervention ladder was applied, confirming least infringement to be the matter of different rationales for the same intervention. Before establishing the ethical justification of the public health intervention, three caveats were suggested: procedural justice, social contexts, and maintenance of trust. Procedural justice, implemented with accountability for reasonableness, can help justify and resolve the issue of infringement by CWF through the fair procedure of four conditions. Considering social contexts and the maintenance of trust can allow CWF to be suitable for the community and transparently communicative for the dental profession, which will be a big challenge and change for the profession in the future. Now that the underlying cognitive dissonance is identified and the caveats for ethical considerations have been offered, further studies are advised about how to reach a consensus on the practical implementation of CWF among relevant stakeholders for the sake of public oral health.

## Figures and Tables

**Figure 1 ijerph-18-02372-f001:**
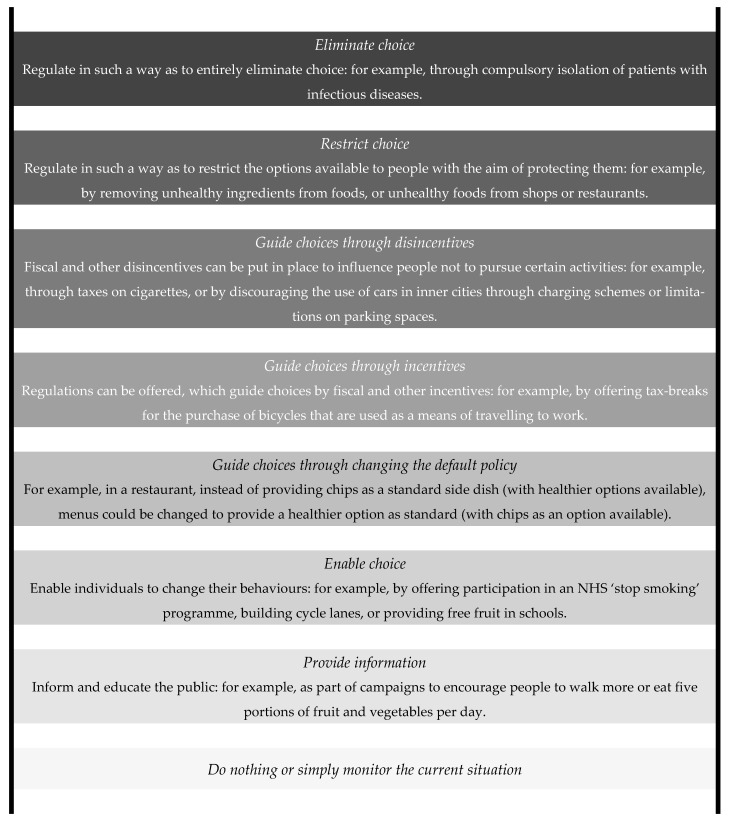
The intervention ladder (reproduced with permission from Nuffield Council on Bioethics) [29] (p. 42).

**Table 1 ijerph-18-02372-t001:** Justificatory conditions [23] and their application to community water fluoridation (CWF).

Justificatory Condition	Meaning	Evaluation of Community Water Fluoridation (CWF)
Effectiveness	Interventions should protect public health effectively.	CWF has been confirmed to reduce dental caries effectively by empirical evidence [3,4,5].
Proportionality	The probable public health benefits should outweigh the infringed moral considerations.	Social and psychological benefits from the decrease in dental caries outweigh the harm of infringed autonomy [26].
Necessity/least infringement	The minimal infringement of moral considerations should have priority among other effective policies.	No other public health intervention can reduce dental caries of the general public as effectively as CWF. Milk fluoridation is also efficient with attached conditions [27] and salt fluoridation may induce hypertension as an adverse effect [25]. Fluoride supplementation holds a higher risk of harmful effects [26] and dental sealants are limited to individual beneficiaries. However, the amount of effects the infringement can affect may differ in accordance with the implementation of CWF.
Public justification	The infringement should be explained and justified to the relevant parties.	Further consideration is required for CWF.

**Table 2 ijerph-18-02372-t002:** Accountability for Reasonableness framework [41] and its application to community water fluoridation (CWF).

Condition	Description	Application to Community Water Fluoridation (CWF)
Publicity	Rationales for decisions should be publicised and approachable.	Principles and evidence for CWF to be publicly accessible.
Relevance	Rationales to provide a *reasonable* construal of how to meet varied health needs of relevant parties.	Reasonable appraisal of the evidence and practice for CWF.
Revision	Mechanisms as a dispute resolution procedure for revisiting decisions in light of counter-arguments and further evidence.	Establishment of the procedure to re-evaluate the implementation of CWF.
Enforcement	Voluntary or public regulation to ensure conditions suggested are satisfied.	Developing a protocol to ensure the fulfillment of three conditions in CWF.

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
