# Peer review of "Community Water Fluoridation: Caveats to Implement Justice in Public Oral Health"

_ijerph, 2021, doi:10.3390/ijerph18052372_

Round 1

Reviewer 1 Report

Community water fluoridation (CWF) is a long-established public oral health intervention with respect to dental caries prevention. However, it remains ethical contention. This work provides a current view of CWF from the perspective of public health ethics and suggests three caveats for further discussion of CWF ethical justification. The topic is interesting and practical. This is a clear, strictly-organized and well-written paper. However, I have a few comments as following:

  1. Two major arguments on ethics of CWF are summarized as principle of non-maleficence and breach of individual autonomy (Line 41-42). How about beneficence and justice?
  2. Utility is one of three principles for public health ethics (Line 98). Why “the utility of CWF is out of focus in the discussion of the paper” (Line 70). I suggest the authors should briefly discuss about this point.

3. Regarding Table 1: What is the modification of this version as compared to the previous work? Why the authors do not separate “necessity” and “least infringement”?

Author Response

Thank you for your considerations. We revised the manuscript carefully based on your recommendations. For the responses, please see the attachment.

Reviewer 2 Report

An very interesting submission that raises a multitude of avenues and reflections necessary to feed a problem that dates back to the 1970s. A few comments and/or elements of personal reflection arising from the reading of this document 

What is a public health problem? Instead of taking Sheiham's reference, my advice would be to take the WHO references

Reflection: Entire regions have natural fluorinated water loaded in excess, is it ethical not to proceed with public health actions of defluoridation?

I fully agree that it is ethical to place ethics beyond the realm of the clinic. Reflection: Can ethics not fluctuate over time in this particular area? A postulate that may have been valid in the 1970s is still relevant in 2020.  In other words, is the community approach to fluoridation still relevant today. The facts show that it is not, except on individuals or groups of individuals. And ethics joins ethics funding.... If we apply Pareto's law, why invest for 100% of the population when 20% investment (individual prophylaxis) covers 80% of the population's needs.

You write (Table 1) "Milk fluoridation is less efficient than CWF [57] and salt". True and false. Milk fluoridation is also efficient (see work from the Borrow i.e. Pakhomov foundation). On the other hand, ethically this procedure has the advantage of substituting a sub-population for a population. This results in the reduction of inequalities because it targets at-risk population groups.
I

n my opinion, there is a dimension missing in your approach which would be, for example: Ethics and politics, therefore health policy. In this precise case it would perhaps have been useful to quote the resolutions (3) of the WHO regarding fluorine. 

A reflection: The profession in the 1950s and beyond, the associated "public health" measures have historically been oriented towards the management or reduction of the incidence and severity of caries. This is still the case today. In other words, our "ethic" is to manage and even validate global policies of companies related to sugar, which is omnipresent and hyper-profitable. And beyond that, to validate the existence and profitability of multinationals involved in the delivery of fluorinated products. 

In your conclusion :
"Considering social contexts and maintenance of trust can allow CWF to be proper for the community and transparently communicative for the dental profession".  My feeling is that the dental profession is in this precise framework not concerned by the problem. It is neither its function nor its responsibility. 

Further studies are advised about how to reach the consensus on CWF for the sake of public oral health: There have already been a multitude of expert committees, how the pursuit of consensus (what consensus?) brings added value.

Author Response

Thank you for your careful examinations. We revised our manuscript based on your suggestions. Please see the attachment.

Round 2

Reviewer 1 Report

No comments